# Porous Nb_2_O_5_ Formed by Anodic Oxidation as the Sulfur Host for Enhanced Performance Lithium-Sulfur Batteries

**DOI:** 10.3390/nano13040777

**Published:** 2023-02-20

**Authors:** Jianming Wang, Lu Chen, Bo Zhao, Chunyong Liang, Hongshui Wang, Yongguang Zhang

**Affiliations:** School of Materials Science and Engineering, Hebei University of Technology, Tianjin 300130, China

**Keywords:** lithium-sulfur battery, niobium pentoxide, porous, cathode

## Abstract

Lithium-sulfur batteries (LSBs), with their high theoretical specific capacity and energy density, have great potential to be a candidate for secondary batteries in the future. However, Li-S batteries suffer from multiple issues and challenges, for example, uneven growth of lithium dendrites, low utilization of the active material (sulfur), and low specific capacity. This paper reports a low-cost and anodic oxidation method to produce niobium pentoxide with a porous structure (P-Nb_2_O_5_). A simple one-step process was used to synthesize P-Nb_2_O_5_ with porous structures by anodizing niobium at 40 V in fluorinated glycerol. The porous Nb_2_O_5_ showed excellent rate capability and good capacity retention by maintaining its structural integrity, allowing us to determine the advantages of its porous structure. As a result of the highly porous structure, the sulfur was not only provided with adequate storage space and abundant adsorption points, but it was also utilized more effectively. The initial discharge capacity with the P-Nb_2_O_5_ cathode rose to 1106.8 mAh·g^−1^ and dropped to 810.7 mAh·g^−1^ after 100 cycles, which demonstrated the good cycling performance of the battery. This work demonstrated that the P-Nb_2_O_5_ prepared by the oxidation method has strong adsorption properties and good chemical affinity.

## 1. Introduction

With the increasing use of electrical energy, LSBs have already been presented in all aspects of everyday life [1]. Compared with traditional insertion lithium-ion batteries, LSBs are promising energy storage systems for the next generation due to their high theoretical capacity (1675 mAh·g^−1^) and high energy density (2600 Wh·kg^−1^) [2,3]. However, the practical implementation of Li-S continues to face formidable challenges, such as the poor electrical conductivity of sulfur, the series of polysulfide intermediates (Li_2_Sn, 6 ≤ *n* ≤ 8) that can dissolve in electrolytes, and the intermediate polysulfides that diffuse to form shuttle effects, which cause pronounced capacity fading [4]. Further, during charging and discharging, the conversion between sulfur and Li_2_S/Li_2_S_2_ can damage the electrode structure and lead to a number of problems such as rapid capacity loss [5]. The performance and the practical application of lithium-sulfur batteries continues to be hindered by these essential problems. A large number of researchers have spent a significant amount of time researching and exploring lithium sulfur battery cathode materials to solve the above problems.

Qiao et al. developed a multistage porous Co-NC@Sn (NH_3_) structure that was constructed as a sulfur host material for Li-S batteries, and it was a simple method for improving Li-S electrochemistry [6]. Manthiram employed a series of materials, i.e., designing and synthesizing a composite using carbon and polar materials, to modify the anode material. For example, mesoporous carbon materials that are well-designed and porous carbon composite materials can provide a large surface area for restraining the polysulfides through physical adsorption [7]. Lu et al. provided three-dimensional (3D) porous graphene foams constructed of reduced graphene oxide (r-GO) for the sulfur cathodes of LSBs. Recently, materials with various pore structure have attracted great attention. The presence of a pore structure can increase the surface energy of a system, which contributes to the improvement of the abundant active sites and ensures the effective wetting and penetration of electrolytes to the electrode surface, facilitating the transfer of interfacial charges [8,9]. A pore structure also provides rapid Li^+^ ion transport highways, resulting in significant rate capabilities [10].

Meanwhile, researchers have spent a significant amount of time and effort studying the use of polar metal oxides in LSBs, and polar metal oxides with strong chemical adsorption, such as TiO_2_, MnO_2_, V_2_O_5_, and Nb_2_O_5_, were investigated to alleviate intermediate polysulfide diffusion (LIPSs, Li_2_Sn, n = 4–8). Zhao et al. developed a V_2_O_3_–VN@NC sulfur host that showed excellent S storage performances, and it had a high reversible capacity, good rate capability, and stable cyclic properties of the reaction process. Using density functional theory computations, they also verified that V_2_O_3_ played a key role in the adsorption of active S during the polysulfide reaction [11]. Song studied a novel reduced graphene oxide (rGO) wrapped a yolk–shell vanadium dioxide (VO_2_) sphere hybrid host (rGO/VO_2_) and determined that the polar VO_2_ could effectively trap polysulfide products by means of strong chemical bonding with oxygen bonds such that the polysulfides largely remained inside the anode electrode material. This designed rGO/VO_2_/S cathode delivered outstanding cycle stability with a low fading rate. Moreover, the porous nature of the conductive 3D porous rGO network facilitated the penetration of the electrolytes and the transportation of the ions, providing efficient transport routes for the lithium ions and electrons and buffering the volume expansion during the lithiation process. Further, this interconnected carbon network could promote chemical interactions with polysulfides. The chemical adsorption of the polysulfide species onto the carbon network hampered the shuttling effect [12].

NbO_x_-based materials (Nb_2_O_5_ and NbO_2_) and their composites have been studied and used for electrode materials for a variety of energy storage systems. They enable a fast recharging, high energy density, and long-term storage [13,14]. Zhan al. synthesized a novel urchin-like Nb_2_O_5_/CNT composite as a modified layer on a polypropylene separator. They prepared precursor materials using hydrothermal synthesis, and then they created the final materials using high-temperature sintering. The synthesized compound material not only provided physical trapping sites for polysulfides, but it also effectively slowed down the diffusion of the polysulfides due to its strong chemical interactions with the polysulfides [15]. Liu al. developed a Pt-Nb_2_O_5_ hybrid catalysis for use in lithium-sulfur battery cathode materials. The synthesized material facilitated the reduction in the higher order polysulfides and the capture of the polysulfides onto the Nb_2_O_5_ [16]. In order to obtain a stronger adsorption capacity of polar materials, the physical morphology factors of polar metal compounds can be studied. An ideal polar compound requires a light weight, a suitable surface for fast Li^+^ transportation, and an adsorptive capacity for S. The reaction equation of a lithium-sulfur battery is as follows: 16Li + S_8_ → 8Li_2_S. The refinement of the reaction process can be divided into the following different stages: (i) S_8_ → Li_2_S_8_, (ii) Li_2_S_8_ → Li_2_S_6_ → Li_2_S_4_, and (iii) Li_2_S_4_ → Li_2_S_2_ → Li_2_S. Preventing the loss of intermediates during a reaction is an effective means of inhibiting the formation of lithium dendrites. It was discovered that Nb_2_O_5_ has high ionic conductivities for Li^+^ ions at the surface, and polar Nb_2_O_5_ also has a strong redox ability for Li_2_S_6_. The redox kinetics of the materials efficiently ameliorated the long-chain shuttle effect of the polysulfides to the Li_2_S_2_/Li_2_S. At the same time, it was found that Nb_2_O_5_ has two forms: crystalline and amorphous. It was also found that the amorphous materials strongly adsorbed lithium polysulfides (Li_2_Sx, x < 6), and we found that although the amorphous materials showed that a remarkable increase in their application had been made possible by constant innovations in the technology of their preparation and their advantages in electrochemistry had increased significantly, the vast majority of catalytic materials that have been developed for sulfur cathodes are crystalline. Therefore, Nb_2_O_5_ material with an amorphous form can be prepared as the cathode mixing material for LSBs, taking into account the advantages of niobium oxide and its amorphous form to improve the overall performance of the battery. However, it was found that Nb_2_O_5_ transformed from an amorphous to a crystalline state after being calcinated at high temperatures. Therefore, a new porous amorphous Nb_2_O_5_ method of preparation was determined to be necessary. Porous amorphous Nb_2_O_5_ was identified as a potential material for lithium-sulfur batteries.

The electrochemical anodization method is a manufacturing process for the preparation of multistage porous oxide films on substrates [17]. The anodization of niobium has been widely studied at the basic research level [18]. In the present work, we introduced P-Nb_2_O_5_ with an amorphous form by means of anodic oxidation to obtain a porous structure [19]. We prepared the material in one step and did not use methods such as hydrothermal synthesis or high temperature calcination. The method used in the experiments was simple and safe, and at the same time, it allowed for the direct preparation of metal oxides with amorphous porous structures. The electrolyte diffusion and penetration were enhanced by the 3D interconnected mesoporous space, and electron transfer kinetics could also be improved by the structures, mitigating the cathode volume change through physical confinement. These porous materials mitigated the shuttle effect through physical trapping within their large surface areas, leading to the high-performance achieved by the LSBs [20].

## 2. Experimental Section

### 2.1. Synthesis of the P-Nb_2_O_5_ Samples

The metal niobium sheet (99.9%, Crown Thailand, Tianjin, China) with a thickness of 0.05 mm was cut to a size of 10 cm × 10 cm. Then, the niobium pieces were sonicated in ethanol solution at room temperature for 30 min and dried in an oven. The dried samples were anodized at 40 V for 2.5 h in 50 mL mixtures of deionized water and glycerol (1:20) containing 0.741 g NH_4_F. During electrochemical oxidation, a graphite electrode was attached to the anode of a Sorensen DLM 300-2 and a niobium metal electrode was attached to the cathode, and the output voltage of the power supply was 40 V. Potentials of 40 V were employed to stimulate oxide development, and anodization was carried out for 150 min. The prepared P-Nb_2_O_5_ was stripped from the matrix with an ultrasonic machine in ethanol, and the oxide layer was collected through centrifugation at 1000 RPM for 10 min. Finally, it was dried at 60 °C in an oven.

### 2.2. Characterization of the P-Nb_2_O_5_ Samples

The structures and shapes of the samples were examined using scanning electron microscopes (SEM, Sigma 500, Oberkochen, Germany), which were equipped with energy dispersive Xray spectroscopy (EDS) systems, and transmission electron microscopes (TEM, JEM-2011) with EDS. Characterization of the morphological and dimensional structures of the samples was completed using atomic force electron microscopy (AFM, Bruker Dimension ICON). The powder XRD patterns were identified by powder X-ray diffraction (Bruker D8 Advance X-ray diffractometer, Cu KR radiation) [21]. Thermal gravity analysis (TGA, Perkin Elmer, Series7, Waltham, MA, USA) was performed to test for sulfur content. A photoelectron spectrometer (ESCALab MKII) was used to perform the X-ray photoelectron spectroscopy (XPS) [22]. Raman spectroscopy was conducted to study the structural and molecular interactions (633 nm, Horiba Jobin Yvon, Stow, MA, USA).

### 2.3. Electrode Prepazration and Battery Fabrication

The batteries with the P-Nb_2_O_5_ mixed positive electrode for Li-S were tested with CR2032-type coin batteries, which were assembled in an Ar-filled glove box (<5 ppm of H_2_O and O_2_). The prepared P-Nb_2_O_5_ material and sublimated sulfur were ground and mixed well in a ratio of 1:3, and then the melt diffusion method was used to further mix them. The mixed material was placed in a reactor under heating conditions of 155 °C for 12 h, and after natural cooling, the final mixed S@P-Nb_2_O_5_ material was obtained. A conventional slurry-coating process was used to fabricate the P-Nb_2_O_5_ electrodes. Subsequently, we prepared N-methyl-2-pyrrolidone (NMP) slurries with 70 wt% active materials, carbon black (20 wt%), and a polyvinylidene fluoride (PVDF) (10 wt%) binder. The amount of sulfur mass in each electrode was approximately 1 mg·cm^−2^. The batteries were assembled in a glove box filled with argon, and the electrochemical tests were conducted on standard CR2032-type coin batteries. S@Nb_2_O_5_ was used as the cathode, Li foil was used as the anode, and 1 M lithium bis (trifluoromethane sulfonel) imide (LiTFSI, 2 wt% LiNO_3_, 1,3-dioxolane DOL, and 1,2-dimethoxyethane DME (*v*/*v*, 1:1)) solution was used as the electrolyte solution. We also tested the purchased granular Nb_2_O_5_ (G-Nb_2_O_5_, 99.9%, Aladdin, Shanghai, China) as a control for the experiment. An electrochemical evaluation was performed using an electrochemical workstation (CHI 660E, Shanghai Chenhua, Shanghai, China) in the potential range of 1.7–2.8 V. Cyclic voltammetry (CV, with a scan rate of 0.05 mV·s^−1^) was used to measure the potential range of 1.7–2.8 V and the impedance spectrum was measured over a frequency range of 100 kHz to 0.01 Hz. The sulfur load and electrolyte doses were 0.5–1.2 mg·cm^−2^ and 40–60 μL, respectively. Battery performance was measured using a Neware BTS4000 battery testing system (Shenzhen, China) at room temperature. In this study, our current density and specific capacity were calculated using the mass of elemental sulfur.

## 3. Results

The preparation of the porous mixed cathode material was realized using the anodic oxidation technique and a subsequent grinding process. The anodic oxidation technique had the advantage of being a more convenient operation method and oxidation process without high-voltage currents, which made it a with low-risk option. The porous structure of P-Nb_2_O_5_ offered a high immobilization ability towards the intermediate polysulfide in the anode material during the oxidation process in the inorganic environment. At the same time, the P-Nb_2_O_5_ microstructure exhibited the presence of abundant channels for rapid ion transport, and it had enough surface area for the reaction energy storage.

The morphology and microstructure of the P-Nb_2_O_5_ sheet were characterized by scanning electron microscopy (SEM) and an atomic force microscope (AFM). The SEM (Figure 1a–c) and AFM (Figure 1d–f) observations showed uniform nanopores of approximately 50 nm in size [23]. The magnified AFM image (Figure 1e) revealed that the aperture depth of P-Nb_2_O_5_ could reach 10–20 μm, and the nanopores were approximately 25–50 nm in diameter. The sulfur could be immobilized by these nanopores during the constant current charge/discharge cycle of the battery [24]. A comparison of the SEM (Appendix A) and AFM (Appendix A) of the unoxidized pure niobium flakes further demonstrated the porous structures formed by the anodic oxidation. As seen in Figure 1f, many pores were connected one another, resulting in the accumulation of abundant pores. The porous structures may have endowed the as-synthesized P-Nb_2_O_5_ material with a high surface area, and this also increased the surface area of the catalyst, which, in turn, could facilitate ion transfer and improve its rate performance. We used G-Nb_2_O_5_ in granular form (Appendix A) the main control for comparison. Further, the energy dispersive spectrometer (EDS) mapping results demonstrated the uniform distribution of the elements (Nb and O) in the oxide (Figure 1g–i), indicating that the Nb_2_O_5_ was successfully prepared. P-Nb_2_O_5_ was mixed with sulfur and then morphologically characterized (Appendix A), and the porous structure was homogeneously mixed with sulfur. Appendix A shows that the synthesized substance had a homogeneous sulfur and P-Nb_2_O_5_ mixture. The overall structure of the material was found to be relatively stable, as indicated by its morphology after cycling (Appendix A.

The greater surface area of the P-Nb_2_O_5_ indicated that there was increased contact area with the active substances. The high-resolution transmission electron microscopy (HRTEM) image showed the P-Nb_2_O_5_ microstructure and a state of void distribution (Figure 2a–c). Moreover, an abundance of mesopores was observed in the oxidized structure of the P-Nb_2_O_5_, which resulted in an increase in the active interfacial energy and the strength of the physical adsorption on the sulfur (Figure 2b). A deeper probe into the amorphous structure was carried out by HRTEM observation. As shown in Figure 2d–f, the energy dispersive spectroscopy (EDS) mapping further demonstrated that that Nb and O were evenly distributed in the oxide (Figure 2e,f), which also indicated the successful synthesis of P-Nb_2_O_5_.

Figure 3a shows the XRD pattern of the P-Nb_2_O_5_, and no diffraction peaks could be detected in the Nb_2_O_5_ pattern, indicating that the synthesized P-Nb_2_O_5_ was amorphous in nature [25]. It also shows that the G-Nb_2_O_5_ was in a crystalline state. The presence of elemental sulfur in the mixture could be determined after mixing the sulfur, as seen in the analysis of the XRD pattern shown in Appendix A. The textural properties of the P-Nb_2_O_5_ with a thickness of approximately 20 μm were characterized by the N_2_ physical adsorption/desorption experiments (Figure 3b,c) [26]. Figure 3b shows the N_2_ adsorption-desorption isotherm of the P-Nb_2_O_5_. The results corresponded to the TEM plots, indicating the presence of many gaps within the P-Nb_2_O_5_. BET analysis determined the specific surface area of the P-Nb_2_O_5_ to be 19.38 m^2^·g^−1^. Pore distributions in the BJH ranged from 2–10 nm, as shown in Figure 3c. Since the P-Nb_2_O_5_ had a porous structure, at the electrolyte–electrode interface, the high specific surface area increased the contact area, providing plentiful active sites for the sulfur interfacial reactions [27]. In addition, the Raman spectra shown in Figure 3d also showed an obvious single peak and peak width, which also indicated the formation of an amorphous substance. We conducted X-ray photoelectron spectroscopy (XPS), shown in Figure 3e,f, to the study of the composition of the amorphous P-Nb_2_O_5_. Through a peak at 206.4 eV, which was related to the Nb3d5/2, and a peak at 209.3 eV, which was related to the Nb3d3/2, the predominant chemical state in the P-Nb_2_O_5_ amorphous material was found to be Nb(V). Additionally, the O1s XPS results could be fitted to correspond to the two peaks. As seen in Figure 3e, the two elements detected in the Nb_2_O_5_ were O and Nb. The two peaks of the O1s centered at 529.3 and 531.1 eV originated from the Nb-O and Nb-OH, respectively (see Figure 3f).

Thermogravimetric analysis (TGA) toward the composite was carried out under an Ar atmosphere (Figure 4) to confirm the content of the sulfur. The weightlessness of the samples was divided into several different stages, depending on the temperature. The weight loss stage was seen at a low temperature range (under 100 °C) and corresponded to the slight mass loss at approximately 100 °C, which was attributed to the evaporation of water from the rising temperature (which was caused by the adsorption of the pores in the samples and could be neglected). It was then evident, as seen in the figure, that these samples showed weight loss from 180 °C to 350 °C in an inert gas environment. At this stage, the TGA diagrams showed a very rapid weight loss, mainly due to the rapid sublimation of sulfur in the composite, with a weight loss of ~74%. The weight percentage of the S in the composite was analyzed to be ~74 wt% using thermogravimetric analysis. The experimental results showed that the P-Nb_2_O_5_ had a high S loading capacity and sulfur-fixing effect.

We created the Li-S batteries using the P-Nb_2_O_5_ as the mixed material in the positive electrode, and we investigated the electrochemical performance of the prepared active material. The specific capacities calculated in this work were all based on sulfur weight [28]. Figure 5a shows the CV curves from the first to third cycles in a voltage range of 1.7~2.8 V at a scan rate of 0.1 mV s^−1^. During the first cathodic reduction step, two reduction peaks at 2.054 and 2.336 V were observed. During the electrochemical reduction, the soluble long-chain polysulfide form at 2.336 V was reduced to an insoluble short-chain polysulfide, forming another cathodic peak at 2.054 V and indicating that the sulfur underwent a multi-step reduction during the whole reaction process. During the subsequent anodic scan process, a strong oxidation peak could be observed at 2.345 V. During the ensuing electrochemical cycle reaction, the curve maintained a strong oxidation peak and small shoulders of 2.345 V and 2.385 V, respectively. This was caused by the transition of the long-chain polysulfides to short-chain polysulfides and the permeation of the electrolytes during the reaction. Further, the peak position and area of the CV curve remained constant from the first to the third cycles, indicating that the P-Nb_2_O_5_ material had excellent capacity retention [29]. Figure 5b displays the initial charge/discharge profiles of the P-Nb_2_O_5_ at 0.2 C in the voltage window of 1.7–2.8 V. It was clear that the P-Nb_2_O_5_ exhibited a higher capacity of 1106.8 mAh·g^−1^, indicating an increased sulfur utilization due to the excellent porous structure of the P-Nb_2_O_5_ electrode.

The cycling performances of the P-Nb_2_O_5_, G-Nb_2_O_5_, and CNT electrodes at 0.2 c are shown in Figure 5c. The initial discharge capacity of the P-Nb_2_O_5_ was 1106.8 mAh·g^−1^, indicating the good sulfur utilization of the electrode, with a capacity storage rate of 73.14% after 100 cycles. Compared to the granular Nb_2_O_5_ and CNT as the cathode mixing material, the battery that used the P-Nb_2_O_5_ as the mixing material for the positive electrodes showed excellent rate capability (Figure 5d,e) and delivered specific capacities of 1108.6, 1016.4, 892.6, 826.4, 753.4, and 713.4 mAh·g^−1^ at 0.2, 0.5, 1, 2, 3, and 5 C, respectively. The promoted rate capability of the P-Nb_2_O_5_ could be attributed to the porous nature, fast charge transfer, and rapid transport capabilities. When the rate returned to 0.2 C, the specific capacities of the batteries with the P-Nb_2_O_5_, G-Nb_2_O_5_, and CNT forms were 958.85, 655.01, and 870.70 mAh·g^−1^, respectively [30]. Electrochemical impedance spectroscopy on the batteries with the G-Nb_2_O_5_ and P-Nb_2_O_5_ electrodes was carried out, and the differences in the internal resistance were investigated, as shown in Figure 5f. The two Nyquist diagrams of the batteries in Figure 5f have the same shape, including a semicircle involving the charge transfer resistance (Rct) and a sloping line associated with the Li diffusion. The P-Nb_2_O_5_ had a lower charge transfer resistance (Rct = 117.6 Ω) compared to the G-Nb_2_O_5_, indicating that the P-Nb_2_O_5_ had the fastest charge transfer kinetics. To further investigate the status of the potential applications of the P-Nb_2_O_5_ cathode in practice, we explored the long cycle of the P-Nb_2_O_5_ and G-Nb_2_O_5_ cathodes, as seen in Figure 5g. The P-Nb_2_O_5_ cathode maintained a reversible capacity of up to 602 mAh·g^−2^ after 400 cycles at 1 C, with very high stability and discharge capacity compared to the G-Nb_2_O_5_ and CNT, which was primarily due to the redox reaction facilitated by the presence of the porous Nb_2_O_5_ and the suppression of the shuttle effect of the LiPSs. This demonstrated that the porous Nb_2_O_5_ could improve the reaction kinetics during the polysulfide reactions. In order to demonstrate the potential of the P-Nb_2_O_5_ for applications in LSBs, cycle performance tests were carried out on the LSBs with high sulfur loadings. As shown in Figure 5h, the electrode exhibited a high area capacity of 4.8 mAh·cm^−2^ at a high sulfur load of 5.2 mg·cm^−2^.

Adsorption experiments were performed on the polysulfides to verify the strength of the P-Nb_2_O_5_ on the polysulfides. The adsorption capacity of the P-Nb_2_O_5_ on the polysulfide was assessed by the observation of the color of the sample in a photograph (Figure 6a, inset shown). After 12 h, the solution in the bottle containing the P-Nb_2_O_5_ changed to near-transparent, indicating the good adsorption of the material to the polysulfide. The results showed that the P-Nb_2_O_5_ had a stronger ability to anchor the polysulfide than the G-Nb_2_O_5_, thus effectively increasing the cycling capacity of the battery. In addition, the strong interaction between the polysulfide and the P-Nb_2_O_5_ was verified by the low UV-Vis spectral intensity (Figure 6a). The LSV curve for the cathode is shown in Figure 6b, in which the P-Nb_2_O_5_ was more porous than the G-Nb_2_O_5_, which had a higher current response, indicating that the redox kinetics of the LiPSs were significantly higher. We further characterized the catalytic effect of the LiPSs by assembling a P-Nb_2_O_5_ symmetric battery with two identical electrodes and by using cv curves. Several highly overlapping CV curves, as seen in Figure 6c, showed that the electrodes had great reversibility. The redox current response of the P-Nb_2_O_5_ electrode was quite high compared to the G-Nb_2_O_5_, as shown in Figure 6d, indicating that the P-Nb_2_O_5_ could greatly improve the conversion of the LiPSs.

To demonstrate more clearly the electrocatalytic performance of the P-Nb_2_O_5_ in the Li-S batteries, the diffusion rate of the lithium ions (Li^+^) was evaluated by CV at different sweep rates. Further, according to the Randle–Sevick equation (Ip = 2.69 × 10^5^n^1.5^aD^0.5^Li^+^ $D_{lI+}^{0.5}$0.5v^0.5^C_Li_), the diffusion coefficient of the lithium ions could be calculated from the peak current (Ip) and the square root of the scan rate (ν^0.5^), which have a linear relationship [31]. In the equation, Ip, n, a, D_Li+_, C_Li_, and v denote the peak current, number of charge transfers, geometric electrode area, geometric electrode area, concentration of lithium ions in the electrolyte, and scan rate, respectively. Therefore, Ip/v^0.5^ can be expressed as the diffusion capacity of the Li^+^. The slope of the final fit is proportional to the square root of D_Li+_. The slope of the P-Nb_2_O_5_ cathode was found to be higher than that of the G-Nb_2_O_5_ cathode, as shown in Figure 6e–h. The values of D_Li+_ at the P-Nb_2_O_5_ cathode peaks A, B, and C were 0.167, 0.091, and 0.077, respectively, and at the G-Nb_2_O_5_ cathode peaks A, B, and C, they were 0.109, 0.040, and 0.038, respectively, as can be seen in the graph. This demonstrated that the P-Nb_2_O_5_ facilitated the conversion of the polysulfides during the insertion/extraction of the Li^+^.

## 4. Conclusions

In summary, we developed a porous metal oxides positive electrode mixed material for reversible Li-S batteries. Using anodizing methods, the Nb_2_O_5_ apertures of nanoscale size were prepared, and they had abundant polysulfide-retaining capabilities and catalytically active sites. Our experiments demonstrated that using P-Nb_2_O_5_ as a mixing material for lithium-sulfur batteries resulted in a capacity of 602 mAh·g^−1^ after 400 cycles. The research provides an effective and viable approach to the development of high-performance LSBs. The porous Nb_2_O_5_ electrodes exhibited superior cycling stability due to their excellent polar adsorption and retention ability for dissolving intermediate polysulfides, which could be attributed to the influence of the porous structure of the Nb_2_O_5_, which provided higher pore volumes and more active sites for sulfur adaptation. The porous structure also reduced the problem of the volume change during the battery cycling. At same time, Nb_2_O_5_ showed a strong entrapment capacity for Li_2_S_6_, and it observably enhanced the redox kinetics from the long-chain shuttle effect. The amorphous characteristics achieved a high conversion of Li_2_Sx, and high sulfur utilization accompanied the growth of the particulate Li_2_S. The results of this study suggest that the low-cost and anodic oxidation method for producing niobium oxide with uniform pores on its surface has promising potential applications for the creation of a porous positive electrode mixed material for higher capacity Li-S batteries. This research offers practical prospects for the study of porous polar metal oxides. The experimental results also offer broad prospects for the development of amorphous porous structures of Nb_2_O_5_ in LSBs.

## Figures and Tables

**Figure 1 nanomaterials-13-00777-f001:**
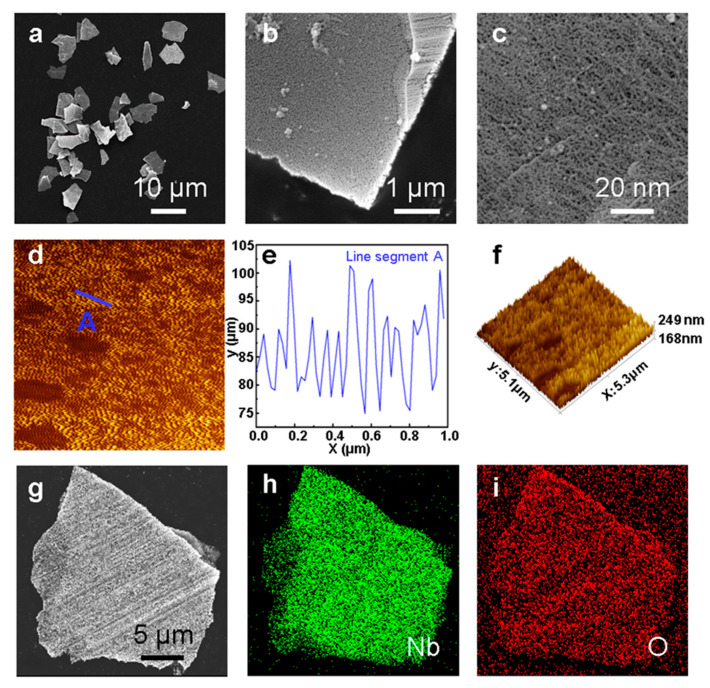
(**a**–**c**) SEM images of the porous niobium oxide obtained by anodic oxidation and (**d**–**f**) AFM images of the porous niobium oxide obtained by anodic oxidation. (**g**–**i**) SEM images and elemental mapping images of P-Nb_2_O_5_. The Nb and O are represented in green and red, respectively.

**Figure 2 nanomaterials-13-00777-f002:**
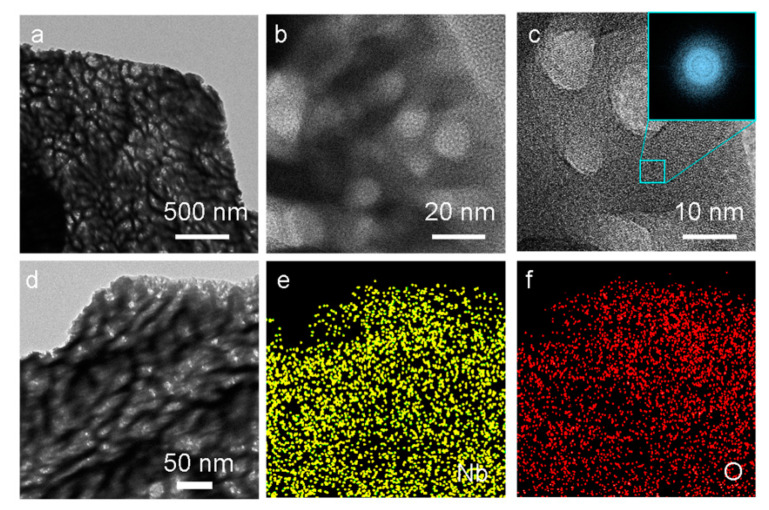
(**a**,**b**) TEM images. (**c**) HRTEM image from the selected area. Inset: fast Fourier transform (FFT) images for framed areas. (**d**–**f**) TEM images and elemental mapping images of the P-Nb_2_O_5_.

**Figure 3 nanomaterials-13-00777-f003:**
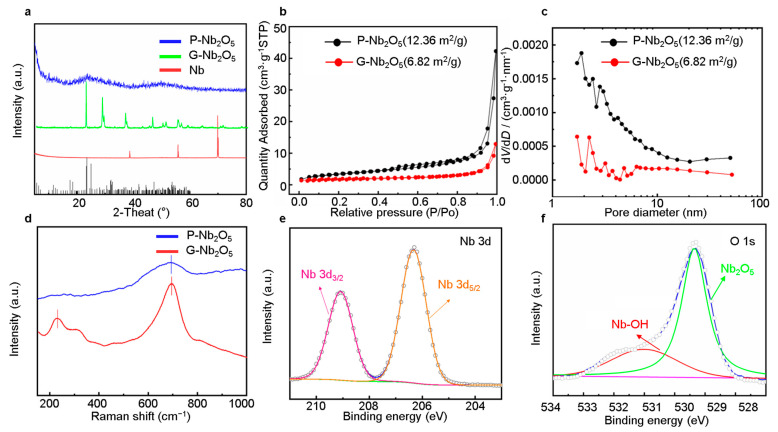
(**a**) XRD patterns; (**b**) adsorption-desorption isotherms of the P-Nb_2_O_5_ and particles of P-Nb_2_O_5_; (**c**) pore size distribution of the P-Nb_2_O_5_ and G-Nb_2_O_5_; (**d**) Raman spectra; (**e**,**f**) high-resolution XPS spectra of the C1s, O1s, and Nb3d peaks of the anodized niobium samples; (**e**) Nb 3d; and (**f**) O1s.

**Figure 4 nanomaterials-13-00777-f004:**
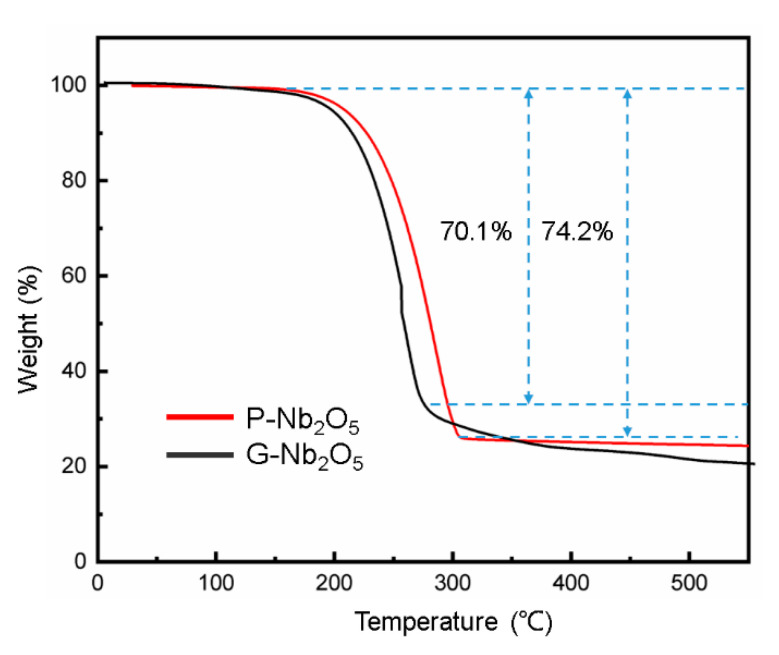
TGA curves of the P-Nb_2_O_5_ and G-Nb_2_O_5_.

**Figure 5 nanomaterials-13-00777-f005:**
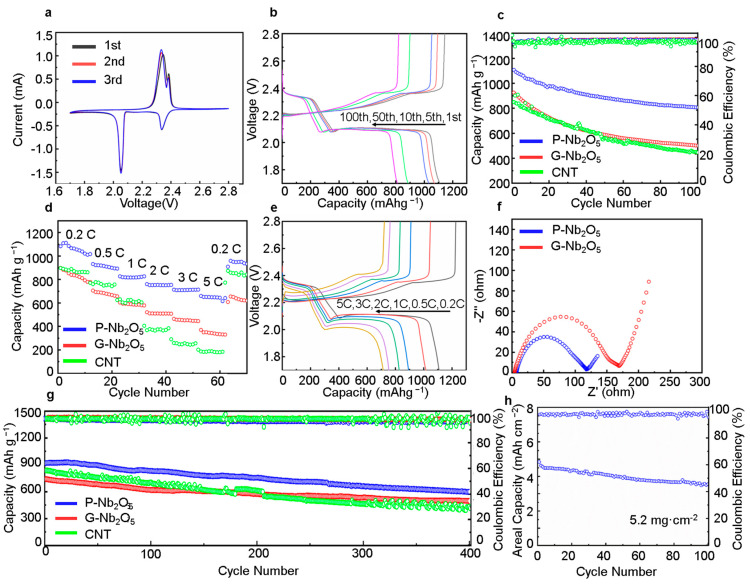
(**a**) CV curves of the batteries with the P-Nb_2_O_5_ cathodes. (**b**) Discharge/charge curves at 0.2 C. (**c**) Cycling performances of the P-Nb_2_O_5_, G-Nb_2_O_5_, and CNT electrodes at 0.2 C for 100 cycles. (**d**) Cycling performances of the P-Nb_2_O_5_, G-Nb_2_O_5_, and CNT electrodes at different current rates (between 0.2 and 0.5). (**e**) Charge/discharge performances at different current rates (between 0.2 and 0.5). (**f**) EIS spectra of the batteries with the P-Nb_2_O_5_ and G-Nb_2_O_5_ electrodes. (**g**) Long-time cycling diagram for the P-Nb_2_O_5_, G-Nb_2_O_5_, and CNT cathodes at 1 C. (**h**) High load cycle performance graph for the P-Nb_2_O_5_.

**Figure 6 nanomaterials-13-00777-f006:**
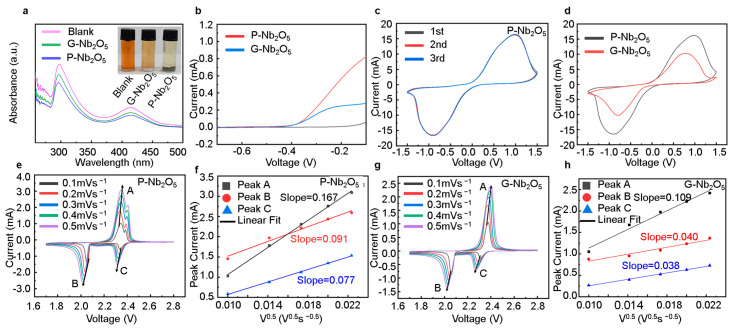
(**a**) UV-vis spectra and optical images of the P-Nb_2_O_5_ and G-Nb_2_O_5_ adsorbed LiPSs solutions. (**b**) Li_2_S oxidation LSV curve. (**c**) Symmetric battery cv curves for the P-Nb_2_O_5_. (**d**) Symmetric battery cv curves for the P-Nb_2_O_5_ and G-Nb_2_O_5_. (**e**,**g**) CV curves at different sweep speeds. (**f**,**h**) Linear regression line for peak A, peak B, and peak C.

## Data Availability

Data are contained within the article.

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
