# Peer review of "Porous Nb2O5 Formed by Anodic Oxidation as the Sulfur Host for Enhanced Performance Lithium-Sulfur Batteries"

_nanomaterials, 2023, doi:10.3390/nano13040777_

Round 1
Reviewer 1 Report
In view of improving the lithium-sulfur batteries' capabilities, the manuscript proposes the use of a porous metal oxides positive electrode obtained through anodic oxidation. The paper is well organized, and the manufacturing stages are described in sufficient detail. Nonetheless, more aspects about the role played by each fabrication step are welcome, in the sense that it would be useful to know how each specific stage influences the final performance of the material with respect to its further utilization. Also, the measurement and test are carefully performed, using state-of-the-art equipment, and the obtained datasets are insightful and meaningful. The following remarks and recommendations are intended to further improve the manuscript.
1. I would suggest increasing the size of Figures 1, 2, 3, 5, and 6 to the allowed width of the page, according to the journal’s template. Many details are rather difficult to observe, and thus can be easily overlooked, at the current size of the figures as they contain multiple subfigures.
2. It would be useful to present the actual electrochemical reactions (equations) in addition to their qualitative descriptions (as text).
3. Al line 23 correct “presented” to “present”.
4. A thorough revision of the English text is needed, both in style and grammar. For example, the paragraph “Besides, this uniquely […] composite materials” contains three unclear phrases. Moreover, the third is exceptionally long (six lines). Also, many phrases begin with “And […]”, a fact that is not very frequent in English, e.g., “And Tian et al. synthesized a novel […]”. Here, “And” can simply be removed.
5. Many words are unnecessarily hyphened in the middle of a line.
6. At line 287, the unit of measurement for the charge transfer resistance Rct is missing.
Author Response
We thank the editor and reviewers for your valuable comments and suggestions. We have studied the reviewers’ comments carefully and highlighted our changes in the revised manuscript.

Reviewer 2 Report
I read the manuscript entitled “Porous Nb2O5 formed by anodic oxidation as sulfur host for 3 enhanced performance lithium sulfur batteries” and have the following suggestions and comments to improve the state of the study.
In the Section 2.2 the following devices are not described AFM, EDS and TGA (only the results of lines 184-190 are shown)
Lines 131, 162, 198: abbreviations DI, DOL and DME, FFT should be written out in full.
In the caption of Fig.2 the authors say "(d-f) TEM images" and in the text they talk about AFM images, please correct.
Where are the Fig 1g-I that the authors discuss in lines 194,195?
Minor points: Line 24 have already presented =have already been presented; Line 231: detect in Nb2O5 =detected in Nb2O5; Line 301: to verified the strength= to verify the strength; Line 194: spec-trometer= spectrometer; Line 196: pre-pared= prepared. Line 187: aper-ture= aperture si alte asemenea greseli de scriere (re-sistance, capaci-ties, ca-pability, cy-cle, facili-tated, poly-sulfide etc.)
Line 253: please correct: “The We fabricated..”
The authors could insert the reference to "the Randles-Sevick equation" discussed
In the conclusions the authors should discuss the (concrete) numerical results obtained from the experiments.
Author Response

(The authors gave the same response as above.)

Reviewer 3 Report
The authors describe a research article entitled “Porous Nb2O5 formed by anodic oxidation as sulfur host for enhanced performance lithium sulfur batteries”. The topic of the manuscript is interesting, and the manuscript constitutes an interesting article concerning the development of porous elecrodes. A short conclusion highlighting the main results of this research article is also provided at the end of the document.
The work is well-written and a well-constructed introduction has been established by the authors. Sufficient spectra and figures are included in the manuscript for comprehension and clarity. Numerous figures in colors have been introduced in the manuscript, rendering the article more attractive. Interesting and convincing results are also presented in this manuscript. Overall, I think that this is a manuscript that I recommend for publication after inclusion of minor revisions.
1) In the conclusion, not so many perspectives are given for this work. Please develop.
2) In the experimental section, reproduction of the anodization reaction in order to prepare P-Nb2O5 samples is not easy, on the basis of the description which is given ate present. Please give more details for reproducibility.
3) Concerning the cycling performance of the batteries, comparison with reference systems should be added in order to evidence the interest of this approach.
Author Response

(The authors gave the same response as above.)

Reviewer 4 Report
The paper by Wang et al. has some strong potentiality; however, I cannot recommend for publication at the present stage for a number of reasons. First of all, English needs extensive revision; many sentences cannot be understood. In particular, the descriptions from lines 184 to 209 are difficult to follow, but there are many points in the paper which are absolutely difficult to read. Moreover, at a certain point, starting from Figure 3 one has a (useful) comparison with a sample called G-Nb2O5. But there is no description of what it is and how it was obtained.
Some other points that the authors should clarify before any resubmission are the following:
- The abstract should be more quantitative and should expose the results of the research
- The introduction could be shortened
- Give information about provider and purity of the starting Nb sheet
- line 158: "PVDF solvent" Are the authors sure?
- line 162: DOL and DME, write the full names before using acronyms
- Figure 3: the surface area of P-Nb2O5 is extremely low. Are the authors sure it can be defined as "porous" or could they use some different expression?
- Fig. 3e: where does Carbon comes from?
- Some SEM, AFM or EDS measurements on samples containing sulphur would be helpful
- Also the conclusions are poorly quantitative
Author Response

(The authors gave the same response as above.)

Reviewer 5 Report
In this study, the authors developed a porous Nb2O5 catalyst that promotes the conversion of sulfur cathode in Li-S batteries. This work introduces an interesting concept. The figures are of appropriate quality. Following the following comments, I recommend accepting this manuscript for publication in Nanomaterials:
1- The following sentence needs to be rewritten in the abstract:
“The polar surface of the P-Nb2O5 can effectively adsorb polysulfides, decreasing the polysulfide shuttling effect, the polar surface exhibits strong chemical absorption for polysulfides, also accelerating redox reaction rates.”
2- What is the meaning of the term "Sulfur fixation" used by the authors? Keywords should be introduced in a more appropriate manner.
3- A number of excellent reviews have highlighted the importance of pores and holes in enhancing electrochemical performance. Authors should explain in detail the importance of structural imperfections in Li-S batteries. The following pioneering reviews should be cited:
Adv. Mater. 32 (2020) 1905923
Progress in Materials Science 116 (2021) 100716
Adv. Energy Mater. 8 (2018) 1702179
4- The text requires careful grammatical corrections before final approval.
5- There have been several publications on the efficient use of Nb2O5 in Li-S batteries. In 2021 and 2022, more than 10 papers were published. It is recommended that authors cite them in the reference list. What is the main difference between this work and previous efforts? The introduction should include a brief summary of recent publications.
6- The introduction is too general, and the cited references fail to provide a detailed discussion of the progress and remaining challenges of Nb2O5-based catalysts for Li-S batteries. It is recommended that authors rewrite this section carefully.
7- In Section 2.1., optical and SEM images of initial Nb sheets should be provided and added to the supplement.
8- Lines 149 and 172: What does the authors mean by “doped”?
9- Line 205: Why did the authors mention heterostructure? There is only one phase of Nb2O5. The text should be corrected.
10- The image in Figures 5a and 6c should be corrected as "3rd". Furthermore, the current densities in Figure 5d should be corrected to 0.2C, 0.5C, 1C, etc.
11- P-Nb2O5/sulfur electrodes with high mass loading should also be tested and reported.
12- Please provide XRD patterns of P-Nb2O5/sulfur and G-Nb2O5/sulfur composites. In addition, it is recommended to present EDS mapping analysis of the Nb, O, and S elements in these composites.
13- The term granular Nb2O5 (G-Nb2O5) should be defined in the first place it was mentioned, not on page 8. Figures 1, 2, and 3 should also include critical characterizations such as XRD, SEM, EDS mapping, and TEM of sample G-Nb2O5.
14- The paper should include additional post-mortem characterizations, such as SEM, XPS, and GITT.
Author Response

(The authors gave the same response as above.)

Round 2
Reviewer 4 Report
The authors sufficiently improved the manuscript. I have just two suggestions:
1) please add the origin of the sample labelled G-Nb2O5;
2) revise English
Author Response

(The authors gave the same response as above.)

Reviewer 5 Report
The authors responded appropriately to my comments. Prior to final acceptance and publication, two minor issues need to be addressed:
1- The authors used a mixture of American and British English in their writing, which should be corrected. For example, both "sulfur" and "sulphur" are used in the text, and we see both "characterization" and "characterisation" words. It is totally unacceptable to use both American and British English in one piece of writing.
2- In Figure 5h, are the capacities 4 mAh/g? In the left vertical axis, the digits or units might be incorrect. It should be corrected.
Author Response

(The authors gave the same response as above.)
